# Molecular Regulatory Mechanisms Drive Emergent Pathogenetic Properties of *Neisseria gonorrhoeae*

**DOI:** 10.3390/microorganisms10050922

**Published:** 2022-04-28

**Authors:** Ashwini Sunkavalli, Ryan McClure, Caroline Genco

**Affiliations:** 1Department of Immunology, Graduate School of Biomedical Sciences, Tufts University School of Medicine, Boston, MA 02111, USA; ashwini.sunkavalli@tufts.edu; 2Pacific Northwest National Laboratory, Richland, WA 99354, USA; ryan.mcclure@pnnl.gov

**Keywords:** regulation of gene expression, *Neisseria gonorrhoeae*, microbiome, microbiome modeling, RNA-sequencing, metal homeostasis, network analysis

## Abstract

*Neisseria gonorrhoeae* is the causative agent of the sexually transmitted infection (STI) gonorrhea, with an estimated 87 million annual cases worldwide. *N. gonorrhoeae* predominantly colonizes the male and female genital tract (FGT). In the FGT, *N. gonorrhoeae* confronts fluctuating levels of nutrients and oxidative and non-oxidative antimicrobial defenses of the immune system, as well as the resident microbiome. One mechanism utilized by *N. gonorrhoeae* to adapt to this dynamic FGT niche is to modulate gene expression primarily through DNA-binding transcriptional regulators. Here, we describe the major *N. gonorrhoeae* transcriptional regulators, genes under their control, and how these regulatory processes lead to pathogenic properties of *N. gonorrhoeae* during natural infection. We also discuss the current knowledge of the structure, function, and diversity of the FGT microbiome and its influence on gonococcal survival and transcriptional responses orchestrated by its DNA-binding regulators. We conclude with recent multi-omics data and modeling tools and their application to FGT microbiome dynamics. Understanding the strategies utilized by *N. gonorrhoeae* to regulate gene expression and their impact on the emergent characteristics of this pathogen during infection has the potential to identify new effective strategies to both treat and prevent gonorrhea.

## 1. Introduction

The Gram-negative bacterium *Neisseria gonorrhoeae* is the causative agent of the sexually transmitted infection (STI) gonorrhea. The geographical distribution of gonorrhea is very diverse. The estimates of gonorrhea cases in adults aged 15 to 49 varied considerably across the different WHO regions. According to WHO statistics, the WHO African region had the highest incidence with 41 cases per 1000 women and 50 cases per 1000 men, whereas the WHO European region had the lowest incidence with seven cases per 1000 women and 11 per 1000 men. A variety of factors contribute to this variation, including sexual orientation, socioeconomic status, demographics, geographical location, cultural practices, and educational background [1,2]. In 2019, 616,392 gonococcal infection cases were reported to the Centers for Disease Control and Prevention (CDC), making it the second most common reportable disease in the United States [3]. A purulent discharge, composed of polymorphonuclear leukocytes (PMNs), exfoliated epithelial cells, and *N. gonorrhoeae*, is a hallmark of gonorrhea in men [4,5,6]. However, subjects infected with *N. gonorrhoeae* are often asymptomatic, an outcome observed more frequently in women than in men [7,8,9,10]. Asymptomatic infections in women are of concern since women are unaware they are infected and do not seek treatment, resulting in prolonged, untreated gonococcal infection. Untreated infections can lead to severe complications, including pelvic inflammatory disease (PID), infertility, and ectopic pregnancy, along with the possibility of further transmission, a public health concern [7,11]. Most gonococcal infections can be successfully treated with oral antibiotic intervention. However, strains of *N. gonorrhoeae* resistant to multiple antibiotics have emerged worldwide [12,13,14,15,16,17,18]. As a result of prolonged and untreated infection and antibiotic-resistant gonococcal strains, gonorrhea remains a significant STI. As such, we need new strategies to combat this infection.

The male and female urogenital tracts are the primary sites of *N. gonorrhoeae* infection, although infections at extragenital sites, including the rectum and pharynx, are sometimes detected [1,19]. The female genital tract (FGT) and its resident microbiome have been proposed to act as a barrier to pathogenic infections [20]. The mucosal immune system of the vagina primarily constitutes an epithelial cell mechanical barrier and innate immune cells, including macrophages and natural killer cells [21,22,23]. Recognition of mucosal pathogens by these immune cells results in the induction and secretion of chemokines, cytokines, innate immune molecules, and the recruitment of PMNs to the site of infection. Innate immune molecules secreted by immune and non-immune cells include antimicrobial peptides (AMPs) that target and disrupt bacterial cell walls, proteins that sequester nutrients essential for pathogens, and reactive oxygen species (ROS) primarily produced by PMNs [21,24,25]. In addition, macrophages, dendritic cells (DCs), and other antigen-presenting cells (APCs) stimulate adaptive immune responses, leading to the initiation of humoral and cellular immunity [21,25]. *N. gonorrhoeae* adapts to this complex FGT mucosal niche through different mechanisms, including regulatory mechanisms that calibrate gene expression in response to different environmental stimuli, such as fluctuating levels of nutrients, antimicrobial compounds, and antibodies directed against various *N. gonorrhoeae* surface antigens. Control of genes involved in these processes is mediated through different mechanisms, such as phase variation and transcriptional regulation through DNA-binding transcriptional regulators and nucleoid-associated proteins (NAP) [26,27,28,29,30,31]. Among other mechanisms, these gene regulatory processes enable *N. gonorrhoeae* to survive in the FGT environment, promoting its pathogenesis [29].

## 2. DNA-Binding Transcriptional Regulators

One mechanism for regulation of gene expression in *N. gonorrhoeae* is achieved through DNA-binding regulators that respond to various stress factors, such as ROS, antibiotics, and nutrient scarcity. This mechanism facilitates the adaptation of *N. gonorrhoeae* to the dynamic environment of the host. Here we describe a subset of the ~34 putative transcriptional regulators identified in *N. gonorrhoeae* based on homology searches, beginning with those responding to varying iron levels at the site of the infection [29].

Iron plays a vital role in various bacterial cellular processes, including DNA replication, transcription, metabolism, and responses to oxidative stress, making it essential for the optimal growth of pathogens [32,33]. Nutritional immunity defense strategies of the human host include depriving pathogens of iron to prevent their survival and pathogenesis [33,34]. This strategy involves maintaining low free iron levels using iron storage proteins including lactoferrin in mucosal tissues and transferrin in plasma, lymph, and cerebrospinal fluid [35,36,37,38]. In the vagina, lactoferrin levels increase in response to bacterial infections and during the mid-follicular phase of the menstrual cycle as estrogen levels rise, further sequestering iron. However, iron from heme becomes more available to bacteria during the menses [35,36]. Despite the importance of iron in numerous biological processes, excess iron may cause ROS generation via the Fenton reaction; therefore, it is crucial for the pathogen to control cellular iron homeostasis during infection [39,40]. In the mucosal niche, gonococci maintain iron homeostasis through a complex iron-responsive network controlled by multiple transcriptional regulators, including the Ferric Uptake Regulator (Fur) and Mtr Protein Efflux Regulator (MpeR) (Figure 1) [29,37,41,42,43,44].

Fur is a global transcriptional iron-response regulator found in Gram-negative and positive bacteria [45]. The classic mechanism of Fur regulation involves Fur binding directly to DNA sequences to inhibit transcription. Without iron, Fur exists as an inactive monomer. However, increasing ferrous iron levels or other divalent cations leads to Fur dimerization and binding to the promoter region of target genes. This interaction blocks subsequent binding by RNA polymerase and leads to decreased transcription of target genes [45,46,47,48]. Fur can also bind to DNA in the absence of iron, a process known as Apo-Fur mediated regulation but this is less common than iron-driven Fur activation [48,49]. In addition to acting as a repressor, Fur can act as a direct activator of gene expression via binding directly to promoter regions, facilitating RNA polymerase binding, and leading to increased transcription of target genes [41,45,50,51]. As a direct regulator of iron homeostasis in *N. gonorrhoeae*, Fur regulates the expression of the genes encoding TonB and TonB-dependent transporters FbpA, TbpA, and TbpB that extract iron from ferritin and transferrin in the host, respectively [52,53,54]. Fur can also indirectly regulate genes by repressing a repressor, with the targets of such repressors transcribed and translated at higher rates. In this indirect role, Fur targets the expression of non-coding RNAs such as regulatory small RNAs (sRNAs), as is the case for the Fur-repressed sRNA NrrF, which controls transcription of the *sdhC/A* genes. Fur-mediated repression of NrrF results in increased translation of *sdhC/A* transcripts; thus, the expression of functional SdhC/A proteins is indirectly activated by Fur [45,55,56,57].

Fur acts as a global transcriptional regulator, regulating the expression of other transcriptional regulators, such as MpeR, another iron response *N. gonorrhoeae* regulator. Under iron-replete conditions, Fur negatively regulates the expression of *mpeR* in an iron-dependent manner [41]. MpeR has two putative α-helix–turn–α-helix (HTH) DNA-binding motifs that share homology with the arabinose operon regulatory protein (AraC) family transcriptional regulators [41,58,59,60]. AraC family regulators participate in multiple cellular processes, including oxidative stress, carbon metabolism, and pathogenesis [59,61]. Like Fur, MpeR functions as a transcriptional activator and repressor. For example, MpeR regulates the *mtrCDE* efflux pump operon by repressing the expression of *mtrR*, the gene encoding the *mtrCDE* efflux pump operon repressor (Figure 1). In contrast to its role as a repressor, MpeR has been shown to activate iron-response genes, including *fetA*, pointing to its role in maintaining the iron homeostasis [42]. The binding of MpeR to the *fetA* promoter has been demonstrated by electrophoretic mobility shift assay (EMSA), but its specific DNA-binding motif has yet to be determined experimentally [42,60]. High-throughput transcriptomic studies of the *N. gonorrhoeae’s* MpeR and H_2_O_2_ regulon have demonstrated that *mpeR* gene expression increases in response to H_2_O_2_ and that MpeR regulates oxidative stress, energy metabolism, and transport genes’ expression [60,62,63]. MpeR also has a role in gonococcal interactions with innate immune cells. *N. gonorrhoeae* upregulates the expression of iron response genes, including *mpeR*, in response to nutritional immune defenses of monocytes [37,64]. Transcriptomic analysis of *N. gonorrhoeae* in cervicovaginal lavage specimens from infected women revealed that *mpeR* is expressed during natural infection by this pathogen [43]. Collectively, these studies show that MpeR responds to different environmental conditions relevant to natural infection, including iron-limitation and ROS (H_2_O_2_), suggesting it is critical in the survival and pathogenesis of *N. gonorrhoeae* during natural infection.

In addition to iron, other trace metals, including zinc and manganese, play a role in the interactions of *N. gonorrhoeae* with the host immune and non-immune cells [29]. Calprotectin, produced by epithelial cells and PMNs, is a zinc and manganese chelator that sequesters these trace metals from pathogens [65]. *N. gonorrhoeae* can extract zinc and manganese from calprotectin via the respective surface-expressed metal transport proteins MntABC and TdfH, whose expression is controlled by the transcriptional regulators Zinc Uptake Regulation Protein (Zur) and Peroxide Responsive Regulator (PerR) [29,66,67]. Furthermore, zinc is also needed for redox homeostasis and is a cofactor of another DNA-binding protein, the redox-responsive regulator NmlR that represses alcohol dehydrogenase (*adhC*) and multi-copper oxidase encoding gene *copA* [68].

Unlike iron which could contribute to ROS generation, manganese protects *N. gonorrhoeae* against oxidative stress by catalyzing ROS removal [66,69,70]. Due to the role of iron and manganese in oxidative stress responses, there is a coordination between iron and manganese homeostasis and ROS defenses [71,72,73]. For example, in *N. gonorrhoeae*, oxidative stress response genes, such as *katA*, are regulated by multiple oxidative-stress response regulators, including OxyR and PerR, as well as iron-response regulator Fur [29,41,62,66,74]. OxyR in *N. gonorrhoeae* negatively regulates the expression of *katA*, with the OxyR regulon consisting of only two additional genes: peroxiredoxin (*prx*) and glutathione reductase (*gor*) [74,75]. This repression of *katA* is contrary to the OxyR in *Escherichia coli* and *Salmonella typhimurium* which regulate *katA* positively [74,75,76]. Another transcriptional regulator involved in the oxidative stress-response is LexA, which represses the expression of three genes, including itself and the DNA-repair enzyme *recN*. Oxidation of a cysteine residue in the LexA protein causes its detachment from DNA, resulting in the de-repression of the three genes in its regulon [77].

As a facultative anaerobe, *N. gonorrhoeae* can be cultured from the female genital tract with obligate anaerobes, and antibodies to proteins required for anaerobiosis have been detected in sera of women with gonorrhea [78,79]. In the absence of oxygen, *N. gonorrhoeae* can use nitrite or nitric oxide as a terminal electron acceptor via the activity of a truncated denitrification pathway composed of a nitrite reductase (*aniA*) and a nitric oxide reductase (*norB*). The expression of genes related to nitrogen respiration is complex as it involves multiple pathways and many proteins that require sensitive transcriptional control. These proteins sense specific molecules and bind to DNA to regulate genes under their control [80,81,82,83,84]. For example, the fumarate nitrate reduction (FNR) protein drives the expression of *aniA* in the absence of oxygen. At the same time, further control is mediated by the ability of NsrR to repress *aniA* expression in the absence of nitric oxide [82]. Additional DNA-binding proteins, including Fur and an ArsR-like protein NsrR, are involved in promoting and repressing the transcription of *norB*, depending on the presence of iron or nitric oxide [41,83,85,86].

## 3. Phase Variation

Phase variation is the reversible, stochastic switching between turning on and off the expression of a gene through genetic or epigenetic mechanisms [87,88]. Genetic regulation of phase variation is dictated by changes in the DNA sequence at a specific genetic locus. Slipped-strand mispairing (SSM) is the most common genetic phase variation mechanism in *Neisseria* spp. and is dictated by different kinds of DNA repeats, including homopolymer tracts and tandem repeats of hundreds of bases in the genome [89,90]. SSM occurs during DNA replication or repair when DNA repeats are misaligned between the mother and daughter DNA strands. DNA repeats are then either expanded or contracted, resulting in phase-variable gene expression at the transcriptional and translational levels, depending on the DNA repeats’ position in reference to its open reading frame [26,87]. Several phase-variable genes encode surface antigens that are potential vaccine candidates [26,91]. Loss or gain in the number of cytosines in the poly-cytosine tract present close to the transcriptional start site of surface-expressed antigen *fetA* determines its expression levels [41,92]. Aside from the insertion of nucleotides, epigenetic regulation of phase variation is determined by the methylation status of the regulatory region of a phase-variable gene or operon rather than their DNA sequences [88,93,94]. Phase variation of DNA methyltransferases leads to coordinated global differential methylation of the bacteria’s genome that corresponds to the on and off methyltransferase variants. Each different global methylation results in the differential expression of a particular gene set called a phasevarion, i.e., phase-variable regulon. Phase variation of the *N. gonorrhoeae* methyltransferase ModA13 comprises 54 genes involved in oxidative stress and antibiotic resistance, including *mtrF* and *trx* [93,94,95].

## 4. Global Gene Co-Expression Network of *N. gonorrhoeae*

Comparative transcriptomics and pathway enrichment analysis of microarray and RNA-Seq (RNA sequencing) data have helped to define global regulons of transcriptional regulators and their associated pathways under specific experimental conditions [41,60,62,81]. However, a systems biology approach can be used to study the interactions of genes from multiple regulons when there is enough transcriptomic data that describes a system (e.g., *N. gonorrhoeae*) under a variety of conditions [96,97]. Recently, the research community has reached this depth of transcriptomic data for *N. gonorrhoeae*, opening up the possibility of gene co-expression analysis of this pathogen [31]. With a gene co-expression network approach, genes represent nodes in a network, and instances of high co-expression between individual gene pairs represent edges in a network. Edges are calculated by analyzing a gene’s expression profile across a range of conditions and linking genes with similar expression profiles (Figure 2) [98]. Using network analysis it is possible to identify new potential targets of known regulators by identifying the edges connecting regulators and their potential targets within the network [31,96]. Network studies have been applied to pathogens such as *Salmonella* to identify processes crucial to the infection [99].

To gain an understanding of the global gene–gene interactions within *N. gonorrhoeae* (including during natural infection and in vitro conditions related to natural infection such as H_2_O_2_ treatment) we constructed the first gene co-expression network for this pathogen (Figure 2) [31]. This network contained 1118 *N. gonorrhoeae* genes (representing 56% of the gonococcal genome) linked by 1499 edges. We then utilized this network to expand our understanding of regulatory pathways within *N. gonorrhoeae*, with a particular focus on Fur. Genes clustered in the network neighborhood of Fur (near Fur in the network) were isolated and further examined. A total of 173 genes were identified in this network neighborhood, including 9 of 23 known targets of Fur. The likelihood of selecting this many Fur targets through random sampling of 173 genes in a network of 1118 was less than 0.002%. We, therefore, reasoned that there might also be undiscovered Fur targets within this network neighborhood. To identify them, we took advantage of the known binding site of Fur in the 5′ UTR of target genes. In our search for known Fur binding sites within the promoter region of genes within the Fur network neighborhood, we identified 11 genes that contain binding sites for Fur but had not been previously identified as Fur targets. In agreement with their role as putative Fur targets, many of these genes are regulated by iron but they lose this regulation in a Fur-knockout strain [31,41]. The discovery of these new potential targets of Fur was only possible after this inference of a gene co-expression network that links, among other genes, regulators, and targets.

We also used a process termed guilty-by-association (GBA) in conjunction with our network to identify putative functions of unknown genes. In a gene co-expression network genes that are linked by edges are likely to participate in similar pathways and share functions. Extending this analysis means that unknown genes can be putatively identified, at least insofar as to what pathways they are involved in, by examining which known characterized genes they associate within the network. This is the basis of the GBA analysis [102]. We applied GBA analysis to our network and found that, for known genes, it assigned the correct function 85% of the time. Applying this approach to all ~700 unknown hypothetical proteins, which also include uncharacterized transcriptional regulators in the *N. gonorrhoeae* genome, we assigned putative functions to 313 (~44%). These functions included gene regulation, DNA metabolism, energy metabolism, general metabolism, phage associated, pilin, replication, stress, translation, and transport-related genes [31].

## 5. Interactions between *N. gonorrhoeae* and the FGT Microbiome

Along with the host immune response, *N. gonorrhoeae* interacts with the resident microbiota of the FGT mucosal niche [29,103]. This community provides additional protection against bacterial, viral, parasitic, and fungal infections. Even in healthy individuals, multiple factors including diet, environment, host genetics, and exposure to microbes during early stages of life contribute to inter- and intrapersonal variability in the microbiome [104,105]. 16S rRNA sequencing analysis is widely used for microbiota profiling; it involves clustering the composite 16S sequences of the microbiota into species, or higher taxonomic levels where species cannot be identified, based on similarities in the sequences [106]. This information is used to measure the diversity metrics alpha and beta diversity that determine the microbiota diversity. Alpha diversity is directly proportional to the number of microbial species and the evenness of their relative abundances within a sample. In contrast, beta diversity is directly proportional to taxonomical differences between samples [104,106,107]. The FGT microbiome encompasses a heterogenous ecosystem with distinct microbiota density and composition at and within the lower and the upper female genital tract (LGT and UGT) [108] The vaginal microbiome at the LGT, consisting of ~200–300 bacterial species, has the lowest species-level alpha diversity, low beta diversity at the genus level compared to the microbiome from other anatomical niches, and high beta diversity at the species level [108,109,110]. Disruption to the composition of the “core healthy microbiota” (dysbiosis) or natural changes in the microbiome due to the changes in female physiology, results in changes to the FGT ecology (Figure 3) [111,112]. In general, bacterial mass at the LGT is several orders of magnitude higher than the UGT, whereas LGT microbiota is less diverse than the UGT [108,110]. *Lactobacillus iners* or *Lactobacillus crispatus* are the predominant microbes in a healthy vagina (LGT), followed by *Lactobacillus jensenii* and *Lactobacillus gasseri*, which are known to play a crucial role in the vaginal microbiome homeostasis [105,108,113]. Non-*lactobacillus* genera including *Gardenella*, *Prevotella*, *Peptoniphilus*, *Peptostreptococcus*, *Anaerococcus*, *Veillonella*, *Megasphaera*, *Leptotrichia*, *Sneathia*, or *Atopobium* represent a small proportion of the healthy vaginal microbiome [108,114]. Unlike the LGT microbiota, the UGT microbiota has traditionally been overlooked because it was considered sterile except when infected with pathogenic bacteria [115]. Additionally, studies on the UGT microbiome are difficult since access to the UGT ecosystem is cumbersome, involving a transcervical procedure risking contamination with cervicovaginal microbiome or invasive methods including hysterectomy and surgical laparoscopy. Nevertheless, recent studies using 16S rRNA amplicon analysis have provided evidence of colonization of distinct microbial communities along the female reproductive tract [109,110,115,116]. Identified major genera of the endometrial microbiota among different studies include *Flavobacterium*, *Gardnerella*, *Bifidobacterium*, *Streptococcus*, *Prevotella*, *Pseudomonas*, *Acinetobacter*, *Vagococcus*, *Sphingobium*, and *Lactobacillus.* It should be noted, however, that these genera do not always appear in all studies, with the exception of *Lactobacillus*, which was consistently observed [109,115,117,118]. The Fallopian tube microbiome is dominated by the phyla Firmicutes (*Staphylococcus* sp., *Enterococcus* sp., and *Lactobacillus* sp.) followed by Pseudomonads (*Pseudomonas* sp. and *Burkholderia* sp.) [115,119,120]. Despite these previous studies, detailed characterization of the UGT microbiome’s structure, function, and diversity is still in its infancy.

Multiple epidemiological studies have shown an association between *Lactobacillus* spp. vaginal occupancy and protection from *N. gonorrhoeae* infections [121]. *Lactobacillus* spp. play a crucial role in maintaining vaginal microbiome homeostasis and providing a barrier against pathogenic microbes by competitive adhesion to the vaginal epithelium and producing antimicrobial products, namely lactic acid and H_2_O_2_ [122,123]. Different strains of *L. crispatus* isolated from the vagina of healthy premenopausal women have been demonstrated to inhibit the growth of *N. gonorrhoeae* in vitro primarily by maintaining an acidic pH from the lactic acid metabolism [124]. *L. jensenii* has been shown to block *N. gonorrhoeae* adherence and invasion of epithelial cells in vitro [125]. Later studies showed *L. jensenii* inhibits *N. gonorrhoeae* adherence to epithelial cells via the production of surface-associated enolase [126]. In *L. crispatus*, enolase and glutamine synthetase [127,128] were shown to interfere with the interaction of *N. gonorrhoeae* with epithelial cells in vitro [128]. In contrast to a healthy *Lactobacillus* population, disruption to the composition of the “core healthy” vaginal microbiota (dysbiosis) is associated with susceptibility to multiple STIs. For example, bacterial vaginosis (BV), a dysbiotic state of the vaginal microbiome, is associated with an increased risk of acquiring STIs, including gonorrhea, chlamydia, HIV, and reproductive and gynecological complications [103,108]. BV is characterized by increased microbiome diversity due to the outgrowth of the anaerobe *Gardnerella vaginalis* along with other anaerobic and facultative organisms including *Atopobium vaginae*, *Bacteroides* spp., *Mobiluncus* spp., and genital mycoplasmas, accompanied by the reduction in *Lactobacillus* spp. Clinical symptoms of BV include (1) vaginal pH > 4.5, (2) grayish-white vaginal discharge with a (3) distinct fishy odor due to the presence of polyamines, and (4) detection of clue cells (epithelial cells covered with Gram-negative rod cells) in the vaginal smears examined under the microscope. Limited studies have examined the mechanistic details of how *N. gonorrhoeae* responds to the healthy and dysbiotic vaginal ecology and how this could potentially impact treatment and vaccine development [129,130,131]. In vitro studies have shown that *N. gonorrhoeae* responds to acidic conditions by modulating the expression of surface-expressed proteins including Rmp, an important determinant of gonococcal vaccine efficacy, and stress response proteins Hsp63 [132,133,134]. Another study has shown *N. gonorrhoeae* can develop resistance to lactic acid in a polyamine-dependent manner [135]. 

Sexual intercourse is associated with changes in the FGT microbiome and the risk of BV in the female partner; this is partly dependent on the composition of the male genital tract (MGT) microbiome [136,137]. Like the FGT microbiome, the MGT microbiome is predominantly present in the LGT, primarily in the urethra and the coronal sack, with variation between individuals. The UGT is typically considered sterile unless infected [138]. A variety of studies have examined the urinary tract microbiome of men without STIs. They have identified considerable diversity among the microbial taxa, including *Corynebacterium*, *Streptococcus*, *Staphylococcus*, *Propionibacterium*, *Sneathia*, *Veillonella*, *Prevotella*, *Ureaplasma*, *Mycoplasma*, *Anaerococcus*, *Atopobium*, *Aerococcus*, *Gemella*, *Enterococcus*, *Finegoldia*, *Lactobacillus*, *Gardnerella*, *Alphaproteobacteria*, and *Prevotella* [139,140,141,142,143,144,145]. Using an approach that incorporated several machine learning classifiers, Mehta et al. have identified penile microbiota in the meatus and the glans/coronal sulcus that accurately predicted the occurrence of BV in a female partner. The following ten meatal taxa are critical for predicting BV incidence, from the most important to the least: *Parvimonas*, *Lactobacillus iners*, *Fastidiosipula*, *Negativicoccus*, *Lactobacillus crispatus*, *Dialister*, *Sneathia sanguinegens*, *Gardnerella vaginalis*, *Prevotella corporis*, and *Corynebacterium*. Notably, some of these bacteria have been associated with BV in the FGT [136]. An analysis of the urethral microbiota of healthy individuals and STI patients found that the amount of *Staphylococcus* spp., *Streptococcus* spp., and *Corynebacterium* spp. was an effective diagnostic indicator for distinguishing the two groups. STI patients had significantly lower levels of the above three species than clinically healthy individuals. On the other hand, the relative number of *Anaerococcus* spp. were significantly higher in men infected with *N. gonorrhoeae* than in healthy men [146].

As with gonorrhea, BV is often asymptomatic and when symptomatic, antibiotics such as metronidazole or clindamycin are commonly used to treat polymicrobial BV, although with a low success rate and subsequent recurrence of symptoms [103,108,147]. Polymicrobial etiology and variations in the BV-associated microbiome between subjects of different racial backgrounds contribute to the low success rate of metronidazole treatment. In some cases, treatment of BV with a combination of antibiotics and *Lactobacillus* probiotics has had a higher success rate than treatment with antibiotics alone [130]. Amstel clinical criteria and Nugent score are the most widely accepted BV diagnostic methods. Nugent score and Amstel’s clinical criteria are subjective diagnostic methods that depend on the observer’s skills to accurately assess the clinical symptoms. This limitation often leads to misdiagnosis [131,148,149]. This subjective diagnosis combined with the fact that many women with BV may be asymptomatic means that it is imperative to use tools that provide a comprehensive picture of the individual-specific BV microbiome in an unbiased manner. A recent study has demonstrated the usefulness of next-generation sequencing for the accurate diagnosis and management of recurrent BV by identifying specific microbes and drug-resistant genes contributing to the symptoms [150]. Other studies have applied metatranscriptomics to identify transcriptionally active microbes, functional pathways, and genes of the vaginal microbiome responsible for resistance to the metronidazole treatment [151,152]. 

## 6. Application of Gene Regulatory Data to Modeling of the Female Genital Tract Microbiome

Network analysis, agent-based modeling, and genome-scale metabolic modeling (GEM) are a few computational analyses used to study various aspects of the microbiome including its structure, function, and dynamics of the microbial community (Table 1). One of the major goals of microbiome modeling is to identify interactions between different microorganisms within a community together with host–microbiome interactions [153]. The inference of such networks and models, and their application to a better understanding of STIs, is based on fundamental gene expression data from *N. gonorrhoeae* and other microbes of the genital tract. Thus, these networks and models serve as links between the gene regulatory pathways described above and the emergent processes of infection. Most of the data collected has come from the global genital tract microbiome; therefore, results have instead focused on the community rather than on the *N. gonorrhoeae*. An analysis of the network provides a structure and function relationship between components of the network based on their connectivity within the bacterial community’s network. It identifies the key microbes or metabolites in the community represented by highly connected hubs/nodes. This network approach is similar to the gene co-expression networks described above. For example, GEM is a data-driven modeling approach that generates a model using experimental genomic or biochemical data [154]. In addition to providing a temporal overview of the bacterial communities, computational modeling can predict how the microbiome will evolve over time [155]. Agent-based modeling uses a stochastic simulation approach by simulating the responses of bacterial communities to perturbations in various biological entities such as an enzyme and its substrate. For example, the flux balance analysis (FBA) approach is used to predict the growth phenotype of a microbe by calculating the flux and flow of metabolites through a network [156].

So far, most modeling work relies exclusively on amplicon analysis. However, the emergent qualities of the vaginal microbiome result from functions expressed by these species. Beyond its role in the diagnosis and management of BV and identifying transcriptionally active microbes, metatranscriptomics analysis of the FGT may be used to model interactions between processes and species in this anatomical site. As a result, other studies have moved beyond amplicon analysis of species and included additional -omics analysis, including metatranscriptomics and metabolomics to better understand how these functions are linked to healthy states [157]. A recent study that examined the Multi-Omics Microbiome Study: Pregnancy Initiative MOMS-PI dataset performed paired metatranscriptomic and metagenomic analyses of 122 vaginal samples (41 from premature births and 81 from term births) and discovered that the taxa that were most transcriptionally active were also associated with premature births. *L. crispatus*, for example, showed some of the highest transcription levels of several genes [158]. Additionally, this study found a higher expression of genes from *G. vaginalis* to be associated with term birth. Another study that examined transcriptomic profiles of vaginal microbiomes from pregnant women also found these higher levels of expression from *G. vaginalis* [159]. The results of these studies indicate that it is the functional pathways expressed by the microbiome that are responsible for its effects on reproductive health. Microbiome and host interactions are likely to lead to these effects and future research should examine this in greater detail.

Modeling of the female genital tract metabolome has also been reported. Noecker et al. used a community-based metabolite potential (CMP) score built from metagenomic data [160]. Each of these scores predicts a community’s potential to deplete or generate each metabolite. These analyses were used to associate specific metabolites and functional pathways to either healthy vaginal microbiomes or those with BV. Metabolomic and taxonomic data from women suffering from vulvovaginal candidiasis (yeast infection) also linked metabolites and microbial taxa and found that *Lactobacillus* abundance was positively associated with lactate and 4-hydroxyphenylacetate, isoleucine, leucine, tryptophan, phenylalanine, and aspartate. *Lactobacillus* was negatively correlated with formate, acetate, 2-hydroxyisovalerate, and alanine. In contrast, other bacterial taxa were positively correlated with the metabolites that *Lactobacillus* was negatively correlated with; these included *Gardnerella*, *Prevotella*, *Megasphaera*, *Atopobium*, *Dialister*, and *Clostridium*. This taxon also showed a positive correlation with organic acids and amines [161]. Further analyses of these types will enable a better understanding of the relationship between taxonomic presence or absence of bacterial species and disease states.

**Table 1 microorganisms-10-00922-t001:** Studies carrying out predictive modeling of the female genital tract.

Study	Input Data (Features)	Modeling	Major Inferences from the Study (Labels)
[162]	A large longitudinal study looking at more than 3620 women with high Nugent scores	Correlative	There is an association between a high Nugent score and acquisition of *N. gonorrhoeae*, *C. trachomatis*, or *T. vaginalis* infection
[163]	16S amplicon data of vaginal swabs from women from four ethnic/racial groups	Correlative	Prediction of *T. vaginalis* infection is associated with high bacterial diversity and reduction in *Lactobacillus* spp.
[158]	An analysis of vaginal samples from women who have experienced preterm or term births (control) using 16S amplicons, metagenomic and metatranscriptomic sequencing was carried out	Associative model using a Mann–Whitney U test and assigning weights to these taxa using L1-regularized logistic regression	The abundance of *Lactobacillus* spp. No difference between pregnant and non-pregnant women Differs in preterm and full-term pregnancies Prediction of preterm birth based on selecting OTUs associated with premature birth Premature birth is significantly associated with four taxa: *Sneathia amnii*, BV-associated bacterium 1 (BVAB1), *Prevotella* cluster 2, and TM7-H1
[164]	16S amplicon timescale data of vaginal samples collected for each subject across 16 weeks	Vagina-specific dynamic microbial interaction network (MIN)	Subject-specific interaction predictions *L. iners* prevents growth of other *Lactobacillus* spp. and *L. jensenii* aids the growth of *Gardnerella* sp. *Finegoldia* sp. have a highly important position in the vaginal microbiome and synergistic relationships with *Sneathia* and *Anarococcus* sp. *L. iners* was found to promote growth of *Gardnerella* as well as to promote growth of *Atopobium*, *Prevotella*, *Parvimonas*, *Sneathia*, and *Mobiluncus*
[114,165]	The longitudinal study included analysis of 16S amplicon sequencing and the Nugent score for vaginal samples	Mixed effects model Dynamic Bayesian network	*L. iners* and *Streptococcus* taxa are linked to menstrual cycle Found positive relationships between *L. iners* and *Atopobium* as well as *Atopobium* and *Gardnerella*
	MOMS-PI dataset metatranscriptomic and metagenomic analysis of 122 vaginal samples		
[160]	Integrated taxonomic and metabolomic data	Community-based metabolite potential (CMP) score	Association of specific metabolites and functional pathways to either healthy vaginal microbiomes or those with BV
[161]	Integrated metabolomic and taxonomic data collected from healthy women and women with BV, vulvovaginal candidiasis, and *Chlamydia trachomatis* infection	Co-abundance network of Spearman correlation coefficient	*Lactobacillus* spp. abundance was positively associated with lactate and 4-hydroxyphenylacetate, isoleucine, leucine, tryptophan, phenylalanine, and aspartate *Lactobacillus* was negatively correlated with formate, acetate, 2-hydroxyisovalerate, and alanine In contrast, other bacterial taxa were positively correlated with the metabolites that *Lactobacillus* was negatively correlated with; these include *Gardnerella*, *Prevotella*, *Megasphaera*, *Atopobium*, *Dialister*, and *Clostridium*. These taxa also showed a positive correlation with organic acids and amines

## 7. Conclusions

As an obligate human pathogen, *N. gonorrhoeae* has evolved mechanisms to colonize the FGT and co-habit with the microbiome, while also being able to adapt to the host immune response. Like other pathogens, *N. gonorrhoeae* has several transcriptional regulators that enable the organism to respond to and survive specific defenses of the host innate immune responses. While we have an overall understanding of how some of these transcriptional regulators function in gene regulation, there are many other uncharacterized putative transcriptional regulators that must be studied. Furthermore, the detailed analysis of gene expression during natural infection is still in its infancy. Moving forward, a major focus should be the integration of molecular-level data collected in vitro with system-level information collected during natural infection. Future studies should also focus on bridging this gap by utilizing various microbiological techniques, RNA-Seq, ChIP-Seq, and bioinformatics analysis tools, such as gene co-expression network analysis and Bayesian network analysis for causality analysis. The same challenges exist in our understanding of the interactions of *N. gonorrhoeae* with the microbiome during natural infection. There have been numerous epidemiological and in vitro studies emphasizing the necessity of the vaginal microbiome, especially *Lactobacillus* species, to protect against infecting pathogens. However, there is still a gap in our understanding of the interactions between functionally active genes, proteins, and metabolites in the microbiome, infecting pathogens, and the host. Future studies should focus on transcriptional, proteomic, and metabolic profiling to understand the impact of the microbiome on the regulation of *N. gonorrhoeae*’s gene expression that results in infection, and susceptibility to antibiotics and probiotic treatment. Collecting -omics data of these types will help in our overall understanding of gonococcal pathogenesis during natural infection of the human genital tract. 

## Figures and Tables

**Figure 1 microorganisms-10-00922-f001:**
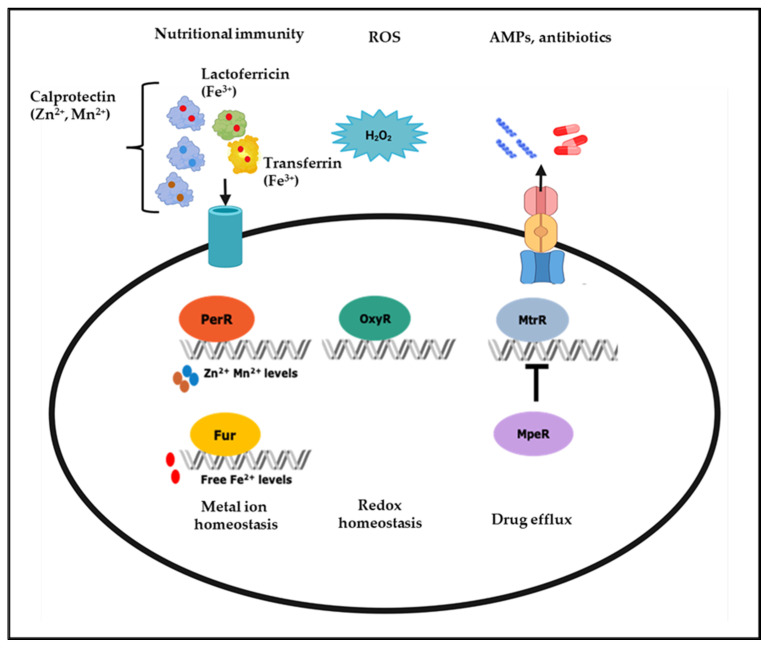
**Transcriptional regulators in *N. gonorrhoeae* respond to various stress stimuli. Nutritional immunity:** Essential trace metals such as Fe^3+^, Zn^2+^, and Mn^2+^ are sequestered from pathogens by being stored in host proteins, such as lactoferricin (Fe^3+^), transferrin (Fe^3+^), and calprotectin (Zn^2+,^ Mn^2+^). The transcriptional regulators Fur and PerR control the expression of genes involved in metal homeostasis, including surface-expressed metal transport proteins that scavenge metals for *N. gonorrhoeae*. **ROS:** the redox-responsive protein OxyR responds to the ROS and H_2_O_2_, and maintains redox homeostasis by regulating the expression of antioxidant genes. **AMPs and antibiotics**: efflux pumps that remove AMPs and antibiotics are negatively regulated by MtrR, which is itself negatively regulated by a second DNA-binding protein, MpeR, such that MpeR positively regulates efflux pumps indirectly. The figure was created with Biorender.com.

**Figure 2 microorganisms-10-00922-f002:**
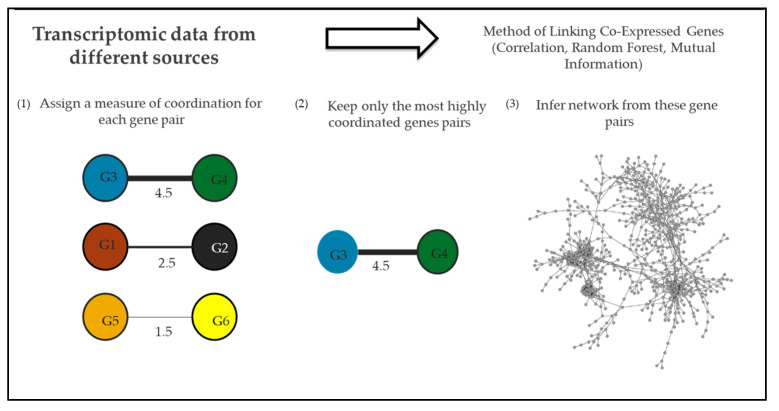
**Network analysis.** To infer a gene co-expression network of a biological system transcriptomic data is first collected. Next, a network inference tool using correlation coefficient (e.g., Pearson), mutual information (e.g., Context Likelihood of Relatedness), random forest (e.g., GENIE3), or another method is used to calculate a co-expression value for each gene pair. Following this, only gene pairs that are highly co-expressed (either positively or negatively) are retained. Once this co-expression and filtering are done to all gene pairs a network of the most highly co-expressed genes can be inferred [96,100,101].

**Figure 3 microorganisms-10-00922-f003:**
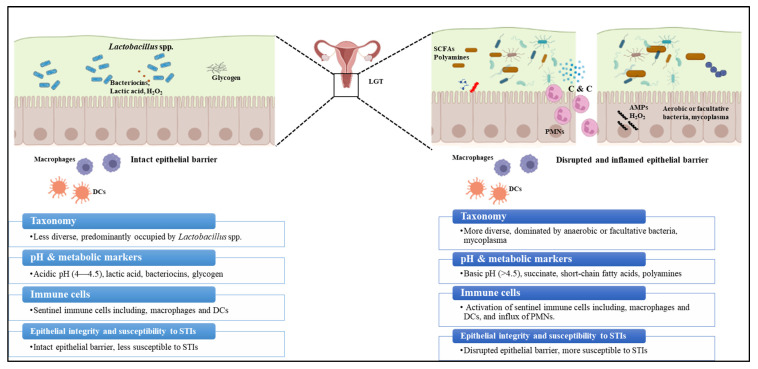
**Impact of a healthy vs dysbiotic microbiome on vaginal ecology. Left Panels:** A healthy vaginal microbiome is less diverse and is dominated by *Lactobacillus* spp. The metabolites of *Lactobacillus* spp. including lactic acid, H_2_O_2_, and bacteriocins that create an environment that is not conducive for the growth of anaerobic bacteria or pathogens. **Right Panels:** Disruption to the core healthy microbiome, as in the case of BV, leads to an environment where *Lactobacillus* spp. is replaced by anaerobic facultative bacteria. This is associated with an increase in pH, SCFAs, and polyamines. Additionally, there is an induction of innate immune responses resulting in upregulation of cytokine and chemokine production, the influx of PMNs, and disrupted epithelial barrier increasing the susceptibility to STIs. **DCs**: dendritic cells; **SCFAs**: short-chain fatty acids; **C & C**: chemokines and cytokines; **AMPs**: antimicrobial peptides; **LGT**: lower genital tract; and **PMNs**: polymorphonuclear leukocytes. This figure was created with Biorender.com.

## Data Availability

Not applicable.

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
