# Peer review of "Molecular Regulatory Mechanisms Drive Emergent Pathogenetic Properties of Neisseria gonorrhoeae"

_microorganisms, 2022, doi:10.3390/microorganisms10050922_

Round 1

Reviewer 1 Report

This is an exciting review on gonococcal transcriptomics and an explanation of the roll of the female microbiome in gonorrhoea. It is excellent narrative review encompassing all the latest insights into the field.

I did not find any glaring errors of omission or any issues with the presentation of the work.

Minor corrections- the entire bibliography has to be cross checked as the species /gene names were not consistently in italics.

Author Response

Comment: Minor corrections- the entire bibliography has to be cross checked as the species /gene names were not consistently in italics.

Response: I appreciate your time and effort in reviewing the paper. The bibliography has been cross-checked, and all the species/gene names have been italicized in the revised document. 

Reviewer 2 Report

This is an excellent and detailed, informative review form Sunkavalli and colleagues on the regulatory mechanisms of Neisseria gonorrhoeae, focused predominantly on the female genital tract. The review is well-written and referenced. I have only a few minor comments.

  1. Line 29; Gc infections are reported as 87 million per year, every year (pandemic!) yet US accounts for only ~600,000. Can the authors provide a small paragraph to describe the global distribution of infections, where they are mostly found.
  2. Interactions of Gc and the GFT microbiome - very interesting, but I am skeptical of the studies of microbiota in the UGT, especially the presence of Pseudomonas, Staphylococcus and Enterococcal spp. This section needs a caveat for these studies and the possibility of contamination.
  3. The article focuses, of course, on the female GT, where much work has been done. But I think the article would be even more informative with a section on the male GT as well. What about the male urethral microbiota? What do we know about this and GC?

Author Response

Thank you so much for reviewing the paper and for the valuable comments you provided. Please find the responses below.

  1. Line 29; Gc infections are reported as 87 million per year, every year (pandemic!) yet US accounts for only ~600,000. Can the authors provide a small paragraph to describe the global distribution of infections, where they are mostly found.
    • Response: This comment has been addressed in the introduction. Please see lines 31-38. 
  2. Interactions of Gc and the GFT microbiome - very interesting, but I am skeptical of the studies of microbiota in the UGT, especially the presence of Pseudomonas, Staphylococcus and Enterococcal spp. This section needs a caveat for these studies and the possibility of contamination.
    • Response: This caveat was brought up in the initial version. Please see lines 300-305 highlighted in red text. 
  3. The article focuses, of course, on the female GT, where much work has been done. But I think the article would be even more informative with a section on the male GT as well. What about the male urethral microbiota? What do we know about this and GC?
    • Response: Male microbiome section is included, please see lines 360-382.